# How reliable is the asset score in measuring socioeconomic status? Comparing asset ownership reported by male and female heads of households

**M. Mahmud Khan**[1]*, **Sebastian Taylor**[2], **Chris Morry**[3], **Shyamkumar Sriram**[4], **Ibrahim Demir**[5], **Mizan Siddiqi**[6]

1 Department of Health Policy and Management, College of Public Health, University of Georgia, Athens, Georgia, United States of America, 2 Global Operations, Royal College of Paediatrics and Child Health, London, United Kingdom, 3 Communication Initiative, British Columbia, Canada, 4 Department of Health Services Policy and Management, University of South Carolina, Columbia, South Carolina, United States of America, 5 Department of Economics, Ankara Yıldırım Beyazıt University, Ankara, Turkey, 6 Vital Strategies, New York, New York, United States of America

* mahmud.khan@uga.edu

**Data Availability Statement:** The relevant variables from the data set are available through the link: (https://osf.io/eyuq2).

## Abstract

Asset scores are widely used as the preferred method of measuring socioeconomic wellbeing of households in developing countries. We examine the degree of discrepancies in reporting asset ownership by male and female heads of the same household. Household asset scores were estimated separately for male and female responses, using Principal Component Analysis, the method widely used in the literature, and households were categorized into wealth quintiles. The results indicate that only half of the households belonged to the same quintile groups for both male and female response-based asset scores. In addition, the two estimates of asset scores within the same quintile deviate by more than 20% for 71% of households in the top three quintiles and for 18% in the poorest two quintiles. Inter-individual (male/female) variability in reporting the asset ownership was high enough to raise concerns about the validity and reliability of asset scores as a metric of household socioeconomic status. Although the study did not try to ascertain underlying reasons for differential reporting, possible explanations could be a lack of awareness among household members on asset ownership or differential propensity to demonstrate relatively better social status of the household by male and female respondents. To improve reliability of asset scores, methodology for collecting asset ownership information should define who in the household may or may not be used as a respondent. Visual verification of reported ownership of assets will reduce male-female discrepancies but the verification process is time-consuming and intrusive, thus negating the advantages of collecting asset data. Alternatives to asset-based scoring need to be considered and one approach could be to solicit subjective opinions from male and female heads on the location of households in the social hierarchy.

**Funding:** This study was funded by the USAID and the Maternal and Child Survival Program. The USAID funding was awarded to Dr. Taylor as the Principal Investigator. The funding agency had no role in designing, analysis and drafting of this specific study.

**Competing interests:** The authors have declared that no competing interests exists.

## 1. Introduction

Over the last two decades, asset scores have become a widely used metric for expressing socio-economic status of households, especially for low-income regions of the world. The Demographic and Health Surveys (DHS) routinely use asset scores to categorize households into socioeconomic groups [1]. More than 400 DHS surveys are now available through the DHS Program website covering about 90 countries [2]. Although household income and expenditure are more direct measures of short-term socioeconomic status, low reliability of the information and high cost of collecting income and expenditure data have led researchers to consider asset ownership-based alternatives [3, 4].

Income and expenditure data are often difficult to collect from households in the developing regions where informal employment remains significant. An evaluation of the Living Standard Measurement Surveys (LSMS) observed that in developing countries expenditure data are usually collected by using "the conventional method of visiting a household once and filling in a detailed schedule covering all the expenditure items." The detailed questionnaire does reduce the effect of memory bias but there is "an upper limit to how detailed a questionnaire may be and there has to be a compromise between the length of the questionnaire and enumerators' and respondents' fatigue. A tired respondent may not give accurate information and this may defeat the purpose of the lengthy schedule" [5]. Another evaluation of income and expenditure data collection methods concluded that "in agricultural/rural economies, home production may account for a significant proportion of a household's consumption. The valuation of such production is a major issue for the calculation of both expenditure and income for households that are both producers and consumers" [6].

To reduce respondent and enumerator fatigue, some surveys ask households to report highly aggregated income or expenditure categories but a comparison of surveys found low consistency between reported aggregate total expenditure and sum of expenditure reported for breakdown items [7, 8]. Male and female household respondents with different access to and experience of employment, labor remuneration and market values may give different estimates of income and expenditure [9, 10]. Even in high income countries, reporting of income was found to be less reliable when the sources of income were from self-employment or financial investments [11]. Moreover, income of households shows strong seasonal variability in rural communities making it difficult to avoid errors in reporting. Sometimes, households may resort to strategic responses when reporting income and expenditure in order to become eligible for means-tested social protection, safety net and/or cash transfer programs.

Several surveys collect data on household income with just a single question covering income from all sources and all earners in the household. Comparing individual respondent's reporting of income with their income in the employer's records in the USA indicates that lower-wage workers overstate their income and high-wage workers under-report income [12]. This review of literature found that misreporting is not only limited to individual and household earned income, but a significant proportion of households also underreport receipt of transfer payments. Without a detailed questionnaire which asks specifically on various sources of income and assets, misreporting appears to be higher [12].

In addition, response or non-response to single question related to household income was found to be non-random–specific characteristics of respondents affected the likelihood of reporting income. For example, increase in the number of adults in the household lowers the probability of reporting. Similarly, women showed lower response rate to the household income question in two surveys in the UK [8]. Clearly, single-question or highly aggregated income category surveys do not provide reliable estimates of household income. Even if we assume that income information can be collected without significant errors, comparing

income across population groups and geographic areas is problematic because of price variability and differences in consumption patterns of households. Therefore, income levels alone will not be a good proxy for socioeconomic status of households.

It is argued that ownership of material assets and access to basic amenities of life should be more reliable due to absence of recall bias (since the questions are related to current ownership of material items) and relative low data needs, avoiding respondent fatigue [13, 14]. Data needs for the calculation of asset-based scores are low, only requiring information on ownership of a limited number of durable assets and access to some basic amenities of life. Filmer et al. [15] found that asset-based socioeconomic indices, calculated using Principal Component Analysis (PCA), are highly correlated with household expenditure in a number of developing countries implying that asset scores can be used as a proxy for representing household expenditure. In fact, they found that the asset score performed better than household expenditure in explaining school enrollment of children in India.

Despite their increasing popularity, a number of concerns have been raised on the use of asset scores as a general measure of socioeconomic wellbeing. First, asset scores probably reflect longer-term socioeconomic status and asset ownership may remain unchanged in the short run even with significant decline in income or expenditure of households [13, 16]. It is also possible that the composition of assets owned by households may change without any change in economic status [6]. Durable assets owned by a household may accrue over time, the so-called asset drift, without any change in the household's underlying wealth [10]. Several studies indicated that the correlations between asset score and household expenditure were not high [17, 18]. Another potential concern is that the assets included in the calculation may not be equally valuable in different regions of a country and inter-regional difference in the utility and economic value of assets may lead to misclassification of households based on asset scores [16, 19]. Therefore, in theory, the larger the geographic area, the asset scores become less reflective of localised or highly-localised differences in asset valuation [20, 21]. Asset scores are not always able to differentiate adequately between 'poor' and 'very poor' households [22], and are not suitable for identifying absolute poverty as it is not designed to reflect households' ability to meet basic needs [23].

The implicit assumption in the use of asset-based measure is that information collected through surveys on durable asset ownership or access to basic amenities of life is highly reliable [16]. Household members are likely to be fully aware of ownership of various durable assets (e.g. ownership of bicycles, TV, car, etc.) and access to amenities of life (source of water, toilet facility, etc.). Unlike the questions on income and expenditures, questions on asset ownership are not cognitively challenging. Asset-based measures also assume that strategic responses in reporting asset ownership will be minimal, if any, as the link between asset ownership and means-tested programs is not clear-cut.

Most surveys collecting household asset information usually ask the questions either to the male head of the household or the female head, or both together in some instances. In general, it is assumed that inter-respondent reliability of reporting assets is relatively high [24]. It is possible, however, that not all assets are equally accessible or utilized by household members. A study in Italy found that men were more consistent in reporting income and employment than women [11]. Misreporting of household ownership of assets or amenities of life may also happen depending upon the sensitivity of the respondent to the level of social status conveyed through the asset ownership. In a multi-country analysis of household wealth surveys, male respondents tended to report a wider distribution of monetary values than female respondents [25].

To understand the degree of bias in reporting asset ownership, this study compared the responses on asset ownership questions by male and female heads of households in northern

Nigeria. The purpose was to examine the degree of correspondence between male and female responses and the effect of differential reporting on categorization of households into wealth quintiles.

## 2. Method

The study used a data set that was collected through household interviews in northern part of Nigeria for understanding the reasons for polio vaccine hesitancy through a funding from USAID. Since the data set collected various information from both the male and female heads of surveyed households, it presented an excellent opportunity to evaluate reliability of asset reporting by comparing male and female responses. In that sense, this data set can be considered "secondary" for the purpose of the study as evaluation of asset scoring was not an objective of the original research. Original research study using the data set was completed in 2016 and the results were published in 2017 [26].

The original study obtained ethical approvals from both the University of East Anglia IRB (where the PI of the project was working at the time of the research) and the Nigerian National Health Research Ethics Board, with the support of an adviser to the National Primary Health Care Development Agency. The field survey followed the interview procedures approved by the IRB. Before conducting the interviews, the interviewers obtained verbal consent from the respondents for their participation in the survey, including separate consents for male and female respondents. The approvals were recorded in the questionnaire and the approval process was observed by the field supervisors. The data collection and analysis followed strict confidentiality protocols, all data were stored and managed on a secure site and all geocodes of household location were removed from the data set prior to the analysis of the data. The names and addresses of households were also removed from the data set prior to sharing the data with co-researchers.

To understand the methodology of evaluating reliability of asset scoring, assume that a survey collects information on asset ownership on $n$ different items. For the $i$th household, the assets owned are shown by the subscript "$o$" and not owned are indicated by the subscripts "$d$". The actual asset ownership status of a hypothetical household is shown by the vector $[A_i]$. The hypothetical vector presented below indicates that the household owns assets 1, 2, 4, etc. but does not own the assets 3, 5, etc.

$$[A_i] = \{W_o^1, W_o^2, W_d^3, W_o^4, W_d^5, \ldots \ldots W_o^n\}$$

If the survey asks a member of the household to report asset ownership, the reported ownership may deviate from the actual for a number of reasons from a purely theoretical perspective. One simple reason could be random error in reporting or recording, which should not be a major concern as averages over an adequate sample size will be representative of underlying asset ownership. It is also possible that not all individuals in the household are fully aware of the specific assets owned, especially for multigeneration households with relatively large size. Another explanation could be that the respondents misreport asset ownership to inflate or deflate household's underlying social status or to avoid taxes. Male and female members are not equally sensitive to the social status conveyed by the reporting of asset ownership and therefore, male and female responses are likely to differ. In one study, husband and wife were asked about the decision-making authority related to specific household actions and the responses differed significantly between male and female heads [27]. A study from Nigeria suggests that male household members had sole custody of household resources, with consequences for differential involvement in decision-making by male and female members [28]. If the benefits of asset ownership affect the wellbeing of male and female members differentially,

the reporting of the assets may become biased as well. Assume that the reported asset ownership vectors for male and female respondents are as shown below. The asset ownership vector reported by male and female respondents are indicated by $A_i^M$ and $A_i^F$ respectively.

$$A_i^M = \{W_d^1, W_o^2, W_o^3, W_d^4, W_o^5, \ldots\ldots W_o^n\}$$

$$A_i^F = \{W_d^1, W_d^2, W_o^3, W_o^4, W_d^5, \ldots\ldots W_d^n\}$$

In this example, the responses on asset ownership by male and female heads are not the same. Although, in theory, reporting of asset ownership may differ based on the gender of respondents, no direct empirical evidence is available to indicate how the discrepancies affect the measurement of overall wealth scores of the households.

This study provides some empirical evidence of differential asset ownership reporting by male and female respondents. The study was designed in 2014 to better understand the attitudes of male and female heads of households towards polio vaccination by assessing various factors, both intra-household and community/environmental, that affect uptake of immunization in northern Nigeria. Since the polio study planned to interview male and female heads of households separately, it presented a perfect opportunity to empirically test the degree of agreement in reporting asset ownership by male and female members of the same household.

During the survey, household heads were asked about their attitudes towards childhood immunizations including propensity to accept polio immunization and at the beginning of the face-to-face survey, respondents were queried about ownership of various assets. In asset-scoring approach, the survey instrument includes questions on access to amenities of life such as sources of drinking water and sanitation facilities. This survey used the same list of assets and amenities the Nigeria Demographic and Health Survey 2013 employed [29]. In most cases, the spouse of the household head was interviewed as the second (or female) head of the household (or 'senior spouse' in the case of polygynous households). The empirical design, requiring interviewing male and female heads of households separately, allowed this study to compare the degree of agreement or disagreement between them in reporting ownership of durable assets and availability of basic amenities of life.

The survey was carried out in three northern states of Nigeria where the incidence of "children missed" during polio immunization campaigns was relatively high. Although the survey targeted a specific geographic location of Nigeria, it should not affect the estimation of male-female differences in reporting asset ownership. Within each state, two Local Government Authorities (LGA) were selected and within each LGA three categories of settlements were selected (administratively defined rural, semi-urban and urban). From the settlements selected, 30 households with young children (less than five years of age) were randomly selected. In some settlements, the field team did not find 30 eligible households and all eligible households in the settlement were interviewed. The survey collected information from 60 settlements and 1,653 households resulting in a total of 3,306 individual interviews using a structured questionnaire. Male and female heads of the households were interviewed separately and by different enumerators ensuring that enumerator bias or bias introduced by the presence of the spouse did not affect the data collection. Female interviewers were employed to collect information from female heads of the households.

Responses to asset ownership questions by male and female heads were compared to measure the degree of agreement or disagreement between the pair of responses. For each asset question, number of male and female heads reporting ownership were obtained. Test of difference of proportions of male and female heads reporting ownership was carried out. The proportions of male and female heads reporting ownership of an asset, however, do not indicate

the degree of mis-match and to understand the male-female mis-match in reporting owner-ship, the degree of disagreement was estimated by the proportions of responses same or not same for male and female heads. To analyze some potential factors affecting male-female dis-agreement in reporting assets, we examined the results by rural and urban locations and degree of religiosity of households. If the differences in reporting is due to intentional misre-porting, one can hypothesize that the degree of religiosity of household members will affect the propensity of misreporting, i.e., the higher is the degree of religiosity, the lower should be the discrepancy. We have calculated an index of religiosity by using information on whether the head of the household attended religious schools, any children attending religious schools and participation in religious discussions and programs in the area in the previous six months. Based on the index, three levels of religiosity were defined: low, medium and high.

Consistent with the methodology used in the literature, Principal Component Analysis (PCA) was carried out to calculate the overall asset scores of households based on the reported asset ownership by male heads and by the female heads separately. The methodology of calcu-lating the wealth index from survey data has been described in a step-by-step manner in a DHS document [30]. In this analysis, we did not correct for rural-urban differences in asset ownership pattern. In the survey area, rural-urban differences were not as pronounced as it was in the national DHS survey. After obtaining the asset scores for the surveyed households, the households were categorized into wealth quintiles based on male and female responses to compare household categorization when male responses are used rather than the female responses and vice-versa.

## 3. Results

The survey collected information from 1,653 Households residing in three northern states of Nigeria, Bauchi, Kano and Sokoto. Some basic characteristics of surveyed households are reported in Table 1 by rural and urban areas. Out of 1,653 households, 542 were located in urban areas and the remaining 1,111 were in rural areas. Male and female heads of these households were successfully interviewed. Number of under-five children per household was about 2.5 for urban areas and 2.7 for rural areas. As expected, less than one percent of house-holds in urban areas while about nine percent in rural areas reported living more than an hour away from the nearest health center. In terms of literacy among female heads interviewed, it is slightly higher in urban areas compared to the rural areas.

The number of male and female heads reporting ownership of various assets and access to amenities of life are presented in Table 2. It also reports the t-statistics for the differences in

**Table 1. Descriptive characteristics of the surveyed households in northern Nigeria.**

| Variable | Urban | Rural | Total |
|---|---|---|---|
| Households (HH) surveyed | 542 | 1,111 | 1,653 |
| Male head interviewed | 542 | 1,111 | 1,653 |
| Female head interviewed | 542 | 1,111 | 1,653 |
| Average number of under-five children per household | 2.52 | 2.72 | 2.65 |
| Percent of HH with 4 or more under-five children | 19.93 | 25.47 | 23.65 |
| Family of household living in the settlement for more than 25 years | 26.38 | 41.67 | 36.67 |
| Percent of HHs located more than an hour away from the nearest health facility | 0.92 | 9.00 | 6.35 |
| Percent of HHs located more than an hour away from the nearest market | 6.46 | 12.24 | 10.34 |
| Percent of female heads literate | 48.52 | 37.08 | 40.83 |

**Table 2. Asset ownership as reported by male and female heads of households surveyed in northern Nigeria, 2014.**

| No. | Type of Assets or amenities of life | Male head reporting ownership | Female head reporting ownership | Both male and female heads reporting ownership | t-statistics, differences in mean, male and female |
|---|---|---|---|---|---|
| 1. | Radio | 1307 | 1244 | 1124 | -2.61 |
| 2. | Television | 469 | 438 | 377 | -1.21 |
| 3. | Iron | 580 | 501 | 386 | -2.93 |
| 4. | Fan | 441 | 405 | 356 | -1.43 |
| 5. | Air Conditioner | 88 | 60 | 28 | -2.36 |
| 6. | Electric Stove | 81 | 72 | 23 | -0.74 |
| 7. | Gas Stove | 231 | 183 | 89 | -2.52 |
| 8. | Kerosene Lamp | 720 | 655 | 479 | -2.29 |
| 9. | Bed | 1550 | 1490 | 1447 | -3.84 |
| 10. | Mattress | 1481 | 1442 | 1379 | -2.12 |
| 11. | Mosquito Net | 1294 | 1290 | 1133 | -0.17 |
| 12. | Refrigerator | 255 | 244 | 161 | -0.53 |
| 13. | Sewing Machine | 296 | 298 | 161 | 0.09 |
| 14. | Table | 491 | 391 | 269 | -3.94 |
| 15. | Sofa | 599 | 458 | 344 | -5.28 |
| 16. | Clock | 788 | 654 | 490 | -4.71 |
| 17. | Generator | 281 | 222 | 155 | -2.86 |
| 18. | Laptop | 72 | 61 | 25 | -0.97 |
| 19. | Telephone | 54 | 33 | 9 | -2.28 |
| 20. | Mobile Phone | 1166 | 1114 | 926 | -1.96 |
| 21. | Motorcycle | 727 | 701 | 494 | -0.91 |
| 22. | Bicycle | 465 | 444 | 264 | -0.82 |
| 23. | Car | 157 | 139 | 76 | -1.1 |
| 24. | Cart | 260 | 263 | 142 | 0.14 |
| 25. | Motorboat | 20 | 19 | 0 | -0.16 |
| 26. | Wheelbarrow | 253 | 270 | 135 | -0.81 |
| 27. | Plough | 420 | 397 | 272 | -0.93 |
| 28. | Agricultural Equipment | 1438 | 1341 | 1242 | -4.62 |
| 29. | Drinking water from piped sources | 101 | 71 | 55 | -1.09 |
| 30. | Drinking water from tubewell, borehole | 32 | 45 | 8 | 2.19 |
| 31. | Drinking water well or spring | 115 | 110 | 67 | -1.18 |
| 32. | Drinking water from other sources | 267 | 315 | 237 | 0.48 |
| 33. | Toilet–Flush latrine (piped to sewer line or septic tank) | 173 | 170 | 89 | -0.49 |
| 34. | Toilet—Pit Latrine | 790 | 767 | 682 | 4.63 |
| 35. | Toilet type–Other | 82 | 82 | 76 | -6.54 |
| 36. | Toilet—No toilet | 62 | 57 | 41 | 2.79 |

[*] Negative t-statistics imply that female heads underreport the item compared to male heads.

t-statistics of 1.96 or higher in absolute terms indicate statistically significant differences between reporting by male and female heads.

mean reporting by female and male heads. The negative value of the t-statistics for an asset imply that higher proportion of male heads reported ownership of the asset than the female heads. In general, male heads tend to report ownership of assets at higher rates than the female heads. The asset types over-reported by more than 20% by male head compared to female

heads were: air conditioner, gas stove, table, sofa, clock, electricity generator and telephone. It appears that a higher proportion of male heads report ownership of high-value assets or durables than the female heads.

To better understand the degree of mis-match between male and female reporting of assets, S1 Annex reports the mis-match. The table shows the number of households in which the male head reported owning an asset but the female head did not, and the number of households where the female head reported owning the asset but male head did not. Combining these two types of errors, the proportion of households with inconsistent reporting of asset ownership by male and female heads are reported. Interestingly, significant over-reporting by male compared to female members is concentrated on some higher-value items, which are also owned by relatively few households overall. Female household heads tend to over-report access to basic amenities of life compared to males. For amenities of life, we can compare the proportions reporting access to "desirable" options for specific amenities of life like water piped into a dwelling or home plot. If we consider the preferred or desirable options, about 20% of households show a mismatch between male and female responses on sources of drinking water and a 26% mismatch for type of latrines used. Therefore, even for access to amenities of life, the degree of discrepancy in reporting was relatively high.

Table 3 reports the degree of mismatch in reporting assets for rural and urban households. The $\chi^2$ values were calculated to test statistical significance of rural-urban differences. The test statistics ($\chi^2$ values) indicate that the male-female mismatch between rural and urban households were not significantly different for household furniture and general household assets taken together but were significantly different for high value assets (e.g., car, electric stove, air conditioner etc.), better quality of amenities of life and ownership of agricultural equipment.

Reporting of household assets by male and female heads were also compared by degree of religiosity of households. Table 4 shows asset ownership reported by male and female heads from households with low, medium and high degree of religiosity. Note that, the degree of mis-match does not appear to be related with religiosity and the discrepancies remain significant even for the high religiosity status of households.

In our survey, since we have two sets of asset ownership responses, one from male heads of households and the other from female heads, asset scores were calculated separately for each. The standard PCA approach was used to derive the differential asset scores, and households were categorized into five equal groups based on those scores. Fig 1 compares the male-head response-based quintile distribution of households for each of the wealth quintiles defined by female head responses. Note that the overlapping of households in the same quintile was not very high. About 58% of households who belonged to the lowest quintile by female-reported asset ownership were categorized as the poorest by wealth scores calculated from male responses. The degree of overlap was found to be lower for quintiles 2–4, ranging from 41% to 45%. For the richest quintile, the overlap between these two sets of household classifications was high, about 76%.

The high degree of concordance of households in the first and fifth quintiles may be due to the definitions of these two quintiles. The lowest quintile (poorest households) is defined by selecting an upper cut-off score while the richest quintile is defined by a lower cut-off. All other quintiles, by definition, must have both the upper and lower cut-offs. Therefore, depending upon the range of the two cut-off values, the degree of overlaps by quintile may become biased. An alternative approach of assessing the degree of divergence or agreement would be to examine the percent of households for which male respondent asset scores deviate from female respondent scores by more than a fixed percent level, say |20| percent. The male-female

**Table 3. Reported ownership of different assets by male and female heads of households in Urban and Rural areas, Northern Nigeria.**

| Asset type | RURAL AREAS | | Percent of households with male-female mismatch in rural areas | URBAN AREAS | | Percent of households with male-female mismatch in urban areas | $\chi^2$ value and df to test the male-female discrepancies for rural and urban households (at 1% level).* |
|---|---|---|---|---|---|---|---|
| | Male-female discrepancies in reporting asset ownership | | | Male-female discrepancies in reporting asset ownership | | | |
| | Male head reporting ownership but not female head | Female head reporting ownership but not male head | | Male head reporting ownership but not female head | Female head reporting ownership but not male head | | |
| **General Household Items** | | | | | | | |
| Radio | 144 | 86 | 20.7 | 39 | 34 | 13.5 | $\chi^2$ = 12.06 < theoretical value of 16.81, df = 6. Rural mismatch not significantly different from urban mismatch |
| Television | 51 | 44 | 8.6 | 41 | 17 | 10.7 | |
| Gas Stove | 76 | 42 | 10.6 | 66 | 52 | 21.8 | |
| Kerosene Lamp | 161 | 109 | 24.3 | 80 | 67 | 27.1 | |
| Iron | 106 | 86 | 17.3 | 88 | 29 | 21.6 | |
| Bicycle | 148 | 120 | 24.1 | 53 | 60 | 20.8 | |
| Fan | 53 | 30 | 7.5 | 32 | 19 | 9.4 | |
| **High value household durables** | | | | | | | |
| Refrigerator | 45 | 36 | 7.3 | 49 | 47 | 17.7 | $\chi^2$ = 38.455 > theoretical value of 21.666, df = 9. Rural mismatch significantly different from urban mismatch |
| Air Conditioner | 10 | 12 | 2 | 50 | 20 | 12.9 | |
| Electric Stove | 15 | 24 | 3.5 | 43 | 25 | 12.5 | |
| Generator | 62 | 40 | 9.2 | 64 | 27 | 16.8 | |
| Laptop | 19 | 15 | 3.1 | 28 | 21 | 9 | |
| Mobile Phone | 164 | 142 | 27.5 | 76 | 46 | 22.5 | |
| Telephone | 28 | 9 | 3.3 | 17 | 15 | 5.9 | |
| Motorcycle | 145 | 137 | 25.4 | 88 | 70 | 29.2 | |
| Car | 35 | 36 | 6.4 | 46 | 27 | 13.5 | |
| Sewing Machine | 78 | 76 | 13.9 | 57 | 61 | 21.8 | |
| **Household Furniture** | | | | | | | |
| Bed | 90 | 25 | 10.4 | 13 | 18 | 5.7 | $\chi^2$ = 14.684 < theoretical value of 15.09, df = 5. Rural mismatch not significantly different from urban mismatch |
| Mattress | 82 | 49 | 11.8 | 20 | 14 | 6.3 | |
| Mosquito Net | 89 | 119 | 18.7 | 72 | 38 | 20.3 | |
| Table | 110 | 89 | 17.9 | 112 | 33 | 26.8 | |
| Sofa | 128 | 88 | 19.4 | 127 | 26 | 28.2 | |
| Clock | 183 | 119 | 27.2 | 115 | 45 | 29.5 | |
| **Agricultural items/ equipment** | | | | | | | |
| Cart | 89 | 93 | 16.4 | 29 | 28 | 10.5 | $\chi^2$ = 19.029 > theoretical value of 13.277, df = 4. Rural mismatch significantly different from urban mismatch |
| Motorboat | 15 | 10 | 2.3 | 5 | 9 | 2.6 | |
| Wheelbarrow | 56 | 77 | 12.0 | 62 | 58 | 22.1 | |
| Plough | 107 | 89 | 17.6 | 41 | 36 | 14.2 | |
| Agricultural Equipment | 96 | 38 | 12.1 | 100 | 61 | 29.7 | |

*(Continued)*

**Table 3.** (Continued)

| Asset type | RURAL AREAS | | Percent of households with male-female mismatch in rural areas | URBAN AREAS | | Percent of households with male-female mismatch in urban areas | χ² value and df to test the male-female discrepancies for rural and urban households (at 1% level).* |
|---|---|---|---|---|---|---|---|
| | Male-female discrepancies in reporting asset ownership | | | Male-female discrepancies in reporting asset ownership | | | |
| | Male head reporting ownership but not female head | Female head reporting ownership but not male head | | Male head reporting ownership but not female head | Female head reporting ownership but not male head | | |
| Water and Sanitation: Amenities of Life | | | | | | | |
| Water: Piped into Dwelling | 5 | 1 | 0.5 | 41 | 15 | 10.3 | χ² = 45.063 > theoretical value of 18.475, df = 7. Rural mismatch significantly different from urban mismatch |
| Water: Piped into yard/plot | 0 | 4 | 0.4 | 24 | 33 | 10.5 | |
| Water: Public Tap | 5 | 7 | 1.1 | 43 | 36 | 14.6 | |
| Water: Tubewell or Borewell | 13 | 40 | 4.8 | 17 | 38 | 10.1 | |
| Water: Protected well | 57 | 52 | 9.8 | 27 | 29 | 10.3 | |
| Flush latrine (piped to sewer) | 1 | 6 | 0.6 | 20 | 10 | 5.5 | |
| Flush latrine (Septic tank) | 6 | 1 | 0.6 | 6 | 10 | 3.0 | |
| Flush latrine (pit latrine) | 11 | 31 | 3.8 | 20 | 38 | 10.7 | |

* χ² values were calculated by considering the differences in mismatch in rural areas compared to urban areas. 1% level of significance was used to conduct the test of similarity, i.e., testing the null hypothesis that rural mismatch is not significantly different from the urban mismatch.

deviation in asset score (D) has been calculated by using the formula:

$$D = \frac{(Asset\ score, male - Asset\ score, female)}{Asset\ score, female} x100.$$

The proportion of households showing more than 20% deviation (positive or negative) between male and female asset ownership was found to be quite high, about 65%. In other words, for 65% of all households in the sample, the asset score derived from male responses deviates from the score for female responses by more than 20%. Even for households categorized in the same quintile groups by these two sets of asset scores, a significant proportion show more than 20% deviation between the two scores. Table 5 reports the proportion of households within the same quintile category whose asset scores deviate by more than 20%. For the poorest two quintiles, about 18% of households in the same quintile group show more than 20% deviation between the scores. For the top two quintiles, about 75% of households in the same quintile category deviate by more than 20%. This implies that even though about half of the households in general overlap by the asset scores, the degree of divergence of household scores remains high, especially for the upper three quintiles. Therefore, the degree of agreement implied by the quintile categories overstates the underlying agreement of asset scores by male and female responses.

**Table 4. Proportions of female and male household heads reporting ownership of selected items by degree of religiosity of the households.**

| Assets | Reported by female household head | | | Reported by male household head | | | % diff M to F | % diff M to F | % diff M to F |
|---|---|---|---|---|---|---|---|---|---|
| | Low religiosity | Medium religiosity | High religiosity | Low religiosity | Medium religiosity | High religiosity | Low religiosity | Medium religiosity | High religiosity |
| Radio | 0.803 | 0.673 | 0.587 | 0.827 | 0.766 | 0.605 | 2.987 | 13.808 | 2.970 |
| TV | 0.311 | 0.220 | 0.058 | 0.336 | 0.214 | 0.087 | 8.000 | -2.564 | 50.000 |
| Iron | 0.330 | 0.270 | 0.192 | 0.387 | 0.318 | 0.180 | 17.204 | 17.708 | -6.061 |
| Fan | 0.282 | 0.208 | 0.081 | 0.301 | 0.234 | 0.110 | 6.940 | 12.162 | 35.714 |
| Air Conditioner | 0.046 | 0.014 | 0.017 | 0.067 | 0.028 | 0.012 | 46.154 | 100.000 | -33.334 |
| Gas Stove | 0.118 | 0.101 | 0.081 | 0.147 | 0.152 | 0.064 | 24.812 | 50.000 | -21.429 |
| Bed | 0.890 | 0.935 | 0.907 | 0.928 | 0.972 | 0.930 | 4.291 | 3.916 | 2.564 |
| Mattress | 0.865 | 0.904 | 0.855 | 0.885 | 0.932 | 0.890 | 2.361 | 3.115 | 4.082 |
| Mosquito net | 0.750 | 0.851 | 0.831 | 0.759 | 0.839 | 0.820 | 1.183 | -1.325 | -1.399 |
| Refrigerator | 0.181 | 0.099 | 0.029 | 0.179 | 0.132 | 0.041 | -1.471 | 34.286 | 40.000 |
| Table | 0.276 | 0.194 | 0.064 | 0.321 | 0.293 | 0.145 | 16.399 | 50.725 | 127.273 |
| Sofa | 0.290 | 0.285 | 0.174 | 0.382 | 0.377 | 0.203 | 31.498 | 32.673 | 16.667 |
| Clock | 0.447 | 0.332 | 0.192 | 0.502 | 0.468 | 0.331 | 12.326 | 40.678 | 72.727 |
| Generator | 0.156 | 0.107 | 0.047 | 0.192 | 0.149 | 0.070 | 22.727 | 39.474 | 50.000 |
| Mobile phone | 0.697 | 0.651 | 0.570 | 0.725 | 0.682 | 0.628 | 3.949 | 4.762 | 10.204 |
| Bicycle | 0.264 | 0.310 | 0.215 | 0.298 | 0.256 | 0.227 | 12.795 | -17.273 | 5.405 |
| Laptop | 0.047 | 0.020 | 0.006 | 0.054 | 0.028 | 0.006 | 15.094 | 42.857 | 0.000 |
| Car | 0.103 | 0.039 | 0.052 | 0.115 | 0.062 | 0.035 | 11.207 | 57.143 | -33.333 |

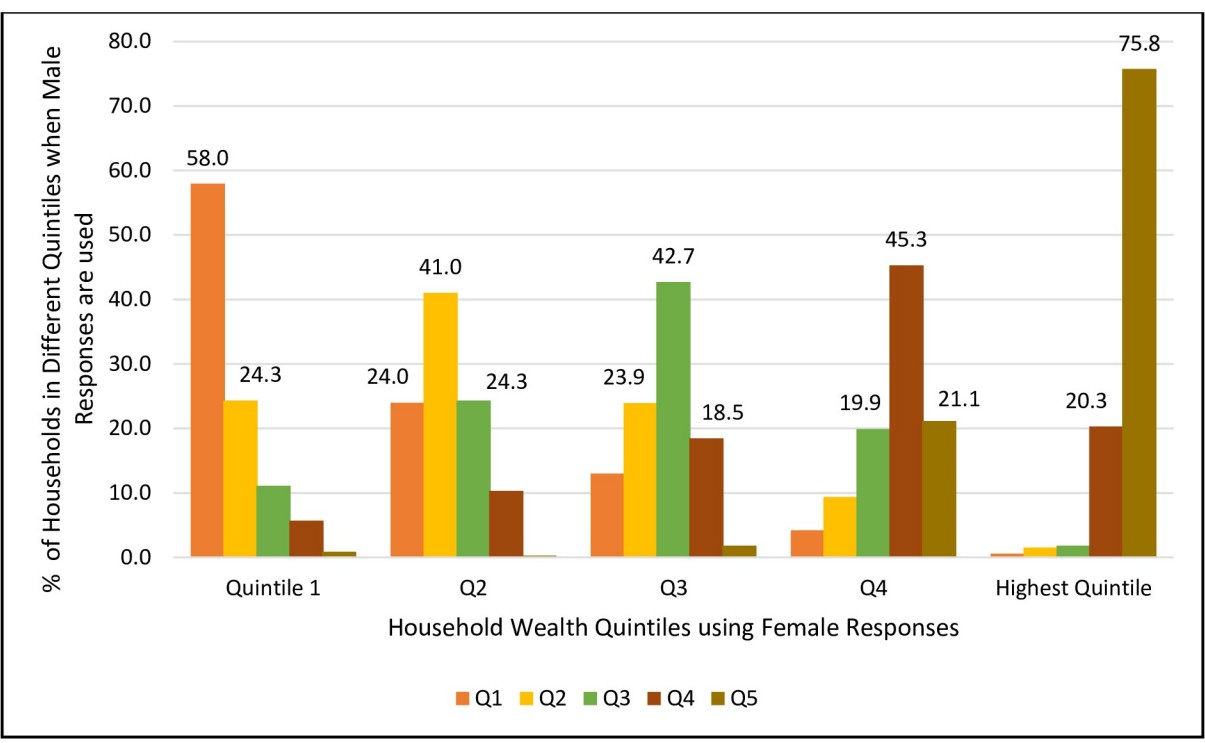

**Fig 1. Distribution of households within each female response-based quintile when male responses are used for asset-score quintile classification.**

**Table 5. Percent of households with the asset scores deviating by more than 20% among the households categorized in the same quintile by both male and female asset scores.**

| Asset score-based quintiles | Number of households in the quintile by both male and female asset scores (%) | Percent of households with asset scores deviating by more than 20% |
|---|---|---|
| Poorest quintile | 193 | 19.2 |
| Second quintile | 135 | 17.0 |
| Third quintile | 141 | 58.9 |
| Fourth quintile | 150 | 84.7 |
| Richest quintile | 250 | 70.0 |

## 4. Discussions

Several studies have demonstrated that asset scoring works quite well as a measure of socioeconomic wellbeing when compared with household income and expenditure-based categorization. This finding has made wealth scoring the preferred measure of socioeconomic wellbeing of households in developing countries. There is, however, a lack of empirical evidence to demonstrate the stability of reported asset ownership in repeated interviews of a household member or interviews of two different household members. In this study, we demonstrate that asset scores calculated from reported asset ownership by male household heads deviate significantly from the scores calculated from female household head responses. In our survey in northern Nigeria, about half of the households were classified in the same quintile groups by the two sets of asset scores. Even within the same quintile group, asset scores deviate by more than 20% for 71% of households in the top three quintiles taken together and for 18% of households in the poorest two quintiles. The lower degree of deviation for the poorest two quintiles is due to lower absolute variation of wealth or asset scores for these quintiles. This is because, households in the poorest quintiles own only a few assets which makes the range of asset scores very narrow for the group.

The findings raise the concern that asset scores may not be as reliable as often assumed. Inter-individual variability of reporting asset ownership (and access to basic amenities of life) can potentially be high. Rural-urban differences in male-female mismatch in reporting assets owned by households makes the problem even more intractable without additional information on how to correct the mismatch. It is not clear why the reported ownership of assets and access to amenities of life differs between household heads. One very straightforward explanation could be that male and female heads are not fully aware of various assets they own and the amenities of life they have access to. Given the wide degree of divergence observed, this explanation is unlikely to be true. Another reason for bias could be differential propensity to demonstrate relatively better social status of household by male and female respondents. Male household heads, in our survey, reported ownership of more expensive assets at a relatively higher rate than female heads. This is consistent with other findings, showing males systematically over-reporting and females under-reporting household resources [9]. This may imply that the male heads are more sensitive to external demonstration of wealth and social wellbeing than the female heads, which may have significant implications both for intra-household decision-making between male and female heads on control, use and allocation of resources, and on social and economic participation outside the household. The relevant question is why do men attempt to convey a better economic status than women? This is perhaps because societies consider it very important for a man to be able to financially support a family to be a good husband and father. Even in developed societies, this perception is widespread [31]. On the other hand, societies consider honesty, friendly, unselfish, sociable, etc., as important qualities of women [32], which may explain the display of humility by women when reporting asset

ownership. One study found that improving economic status tends to benefit male members more than the female members in a traditional society and the work-load of female members remain quite similar across all socioeconomic categories [33]. Therefore, the propensity to demonstrate higher economic status will be lower for female members.

## 5. Conclusions

The asset scores of households vary due to inter-individual variability in reporting asset ownership within the same household and this requires a more nuanced approach to measurement of socioeconomic status. To improve reliability of wealth scoring, researchers should carefully decide who may or may not be considered a possible respondent on asset questions. Visual verification of at least some of the reported asset ownership will reduce the degree of mismatch but visual verification is quite intrusive and time-consuming. In any case, variability in reporting of asset ownership by male and female heads of households imply that the asset scores are not very reliable.

The search for a reliable measure of socioeconomic status should explore the use of subjective assessments of households' status in the community. When determining the subjective ranking of social status, household heads need to compare their own situation in relation to others in the community or the country. MacArthur's Subjective Social Status ladder [34] can be used to test male-female disparity in placing the household in the social hierarchy. If this type of single item composite scale can explain variability of health, educational attainment and economic status of households with lower degree of inter-member disagreement on the location of household in the social ladder, it may turn out to be more reliable than the asset-based scores.

## Supporting information

**S1 Annex. Reported ownership of different assets by male and female heads of households surveyed in northern Nigeria.**
(DOCX)

## Author Contributions

**Conceptualization:** M. Mahmud Khan, Sebastian Taylor, Chris Morry.

**Data curation:** Shyamkumar Sriram, Mizan Siddiqi.

**Formal analysis:** M. Mahmud Khan, Shyamkumar Sriram.

**Funding acquisition:** Sebastian Taylor, Chris Morry.

**Methodology:** M. Mahmud Khan.

**Supervision:** Sebastian Taylor, Mizan Siddiqi.

**Writing – original draft:** M. Mahmud Khan.

**Writing – review & editing:** M. Mahmud Khan, Sebastian Taylor, Chris Morry, Ibrahim Demir.

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
