## [Decision Letter · Decision Letter 0]

2 Sep 2022

PONE-D-22-10525How reliable is the asset score in measuring socioeconomic status? Comparing asset ownership reported by male and female heads of householdsPLOS ONE

Dear Dr. Khan,

Thank you for submitting your manuscript to PLOS ONE. After careful consideration, we feel that it has merit but does not fully meet PLOS ONE’s publication criteria as it currently stands. Therefore, we invite you to submit a revised version of the manuscript that addresses the points raised during the review process.

 Your manuscript has been reviewed by two peer-reviewers and their reports are appended below.  The reviewers comment that your study could be strengthened by improvements to the overall reporting of the study. In addition, the reviewers have commented that the conclusions drawn in this study need to be expanded. Could you please revise the manuscript to carefully address the concerns raised? 

We look forward to receiving your revised manuscript.

Kind regards,

Maria Elisabeth Johanna Zalm, Ph.D

Editorial Office

PLOS ONE

Journal Requirements:

2. In the ethics statement in the Methods, you have specified that verbal consent was obtained. Please provide additional details regarding how this consent was documented and witnessed, and state whether this was approved by the IRB

3. Please include a complete copy of PLOS’ questionnaire on inclusivity in global research in your revised manuscript. Our policy for research in this area aims to improve transparency in the reporting of research performed outside of researchers’ own country or community. The policy applies to researchers who have travelled to a different country to conduct research, research with Indigenous populations or their lands, and research on cultural artefacts. The questionnaire can also be requested at the journal’s discretion for any other submissions, even if these conditions are not met.  Please find more information on the policy and a link to download a blank copy of the questionnaire here: https://journals.plos.org/plosone/s/best-practices-in-research-reporting. Please upload a completed version of your questionnaire as Supporting Information when you resubmit your manuscript.

This study was funded by the USAID and the Maternal and Child Survival Program. The USAID funding was awarded to Dr. Taylor as the Principal Investigator.

Reviewers' comments:

Reviewer's Responses to Questions

**Comments to the Author**

1. Is the manuscript technically sound, and do the data support the conclusions?

Reviewer #1: Partly

Reviewer #2: Yes

2. Has the statistical analysis been performed appropriately and rigorously? 

Reviewer #1: I Don't Know

Reviewer #2: Yes

3. Have the authors made all data underlying the findings in their manuscript fully available?

Reviewer #1: No

Reviewer #2: No

4. Is the manuscript presented in an intelligible fashion and written in standard English?

Reviewer #1: No

Reviewer #2: Yes

5. Review Comments to the Author

Reviewer #1: This is a very interesting paper that provides direct evidence about an assumption made in economics about household asset reporting.

However, the paper is not structured as expected for a PLOS One paper - the usual paper structure is Introduction, Methods, Results, Discussion, and Conclusion which you follow but the contents in each section are mixed up. The following needs to be addressed:

1. In an economics paper the introduction can be longer than in a scientific paper but this one is too long and needs to be tightened up considerably, although it is mostly written for a non-economist audience which is good. The methods section contains some background which belongs in the introduction.

2. The methods section should stand alone and describe all the work undertaken with the data. You do not describe the various measures in sufficient detail. Do not introduce new information as you write up the results - for example combining urban and semi-urban for some analyses and the calculation of the religiosity score. On page 10, the DHS reference is missing in the references

3. Results. It is usual to start with a descriptive table of the study population. It is difficult to determine when you are talking about Table 1 and Annex Table 1 - try combining the two by eliminating some columns. In Table 5, the number of children may be driven more by the length of time the parents have been together. All the tables need p-values where relevant. Why does the number of assets listed decline steadily for each table? I feel there is too much discussion in this section such as the comments about men knowing about water sources.

4. Some tables might be clearer as figures

There is some interesting findings here but they are buried in a plethora of analyses. Strip it back to a more straight forward message that is easier to follow. You don't need to report everything you did.

I think this paper should just be restricted to the assets analysis while the examination of possible explanatory social factors left for another paper. In my experience with risk perception people's behaviours are not consistent across domains and it is unwise to infer this.

Reviewer #2: How reliable is the asset score in measuring socioeconomic status? Comparing asset ownership reported by male and female heads of households

The researchers report on a large-scale study, in which over 1,500 households in northern Nigeria were interviewed, to assess whether there are discrepancies in asset scores between male and female heads of household. They report asset scores (derived from ownership of assets and access to important amenities), a common measure to categorize households into different SES groups, are not reliable, as they report significant asset score differences between M and F HH.

The authors’ research is novel, and addresses an important topic in the literature, which will be of interest to readership. They report on significant discrepancies on asset scores between male and female heads of households in northern Nigeria, underscoring the low reliability of this commonly used measure of SES classification. The manuscript is well written, and their findings are consistent with their research question and methodological approach. One area of concern is in their conclusion - given the low reliability of asset scores, what alternatives are available? I recommend the authors discuss the pitfalls of ‘objective’ measures of SES, and how those might be complemented by ‘subjective’ measures of SES (e.g., subjective social status; one’s perceived position in their own community).

Intro

Discusses issues on variability of income and prestige - could expand on the difficulties in comparing income across populations, further complicating the use of income alone. Making an X amount of money in a rural underserved area allows for significantly different buying power and standard of living in comparison to making the same amount of money in an urban area (where one would likely have less buying power).

Need more references to support the assertion that ‘surveys on durable asset ownership or access to amenities should be reliable’. Based on which theory? Unless you make your assumptions clear, you are neglecting the significant impact of malingering, and how people place value on different things. The durable assets might not be equally valuable across different people (not only inter-regionally, but interpersonally).

The rationale for the manuscript is clear - and I commend the authors for their clear writing. Nevertheless, the justification for comparing the responses of male and female heads is not well-defined. Why pursue this difference? What could account for possible differences in asset evaluation, that would justify your study? Consider including social determinants that could contribute and help justify your study.

Method

Were the enumerators different for the male and female heads of household? Were all female heads interviewed by women, and all male heads by men? Consider clarifying this - and whether gender role expectations could have played a part on the data collection (if so, it is important to note this on the method and the limitation section).

Result

Males over-reported higher-value items and underreported household items, whereas females over-reported basic amenities.

High religiosity and vaccine misinformation associated with higher discrepancy

Table 6 clearly demonstrates the low reliability of asset scores - low reliability for all quintiles (from 40-75%)

Discussion/Conclusion

Individual variability in asset scores, particularly as it relates to male and female heads of household. It appears that impression management (attempts to control how the researcher perceives the participants) is interpreted and acted different upon by males and females HH, which is an important finding. However, the authors did not provide clear theoretical interpretations of these findings - they noted that male and female members might have different levels of awareness of assets and amenities, which albeit true, is unlikely to account for over 50% of discrepancy. The authors touch on this already, but it might be relevant to explain why male HH over-report and female HH under-report household resources. What are the differential social, gender, and cultural expectations placed on M and F HH? Why would M be more likely to boast about their assets, and attempt to look better for researchers? Why would F be more likely to display humility and underreport? What is the benefit of this for the participants?

Overall, the implications of the study are clear and significant. It is clear that there are issues interpreting asset scores, particularly given their low reliability. Nevertheless - the authors did not address whether subjective assessment of one’s social status and position in their community would be a more reliable way of measuring and classifying participants into social class groups (i.e., socioeconomic groups). Although gender roles and different values placed in assets appear to account for significant differences in asset scores, it is paramount to investigate whether the subjective position in their community would be similar between male and female heads of households. For example, using the MacArthur’s Subjective Social Status Scale, which is a visual representation of one’s community in a ladder, with the ones at the top being the ‘best off’ and the ones at the bottom being the ‘worst off’ - where would male and female HH place themselves? I am aware this was not addressed in this research - but I assume that the subjective evaluation of status would be more reliable between male and female HH, as they compare themselves to their surrounding community (and their nation). This is addressed by the researchers - but there is significant sources of error in using objective measures of SES, given within-group variability in income groups in terms of prestige, buying power, worldview, and standard of living. I highly recommend the authors expand their conclusion - visual verification of asset ownership (as suggested) would still be prone to biases from the researcher and the one reporting (i.e., taking them to verify). Same with education - as a simple example, a university professor in the UK, who usually has a doctorate, typically earns less than a plumber, which does not require advanced education but is a highly skilled and highly paid profession. Education by itself also will not provide an accurate picture of status and SES - a combination of asset scores and subjective social status would likely be better comprehensive picture. Consider including a discussion on subjective social status, and how it can serve as a way to address the reliability issues of asset scores alone.

6. PLOS authors have the option to publish the peer review history of their article (what does this mean?). If published, this will include your full peer review and any attached files.

Reviewer #1: **Yes: **Professor Shona Kelly

Reviewer #2: **Yes: **Klaus Cavalhieri, Ph.D.

---

## [Author Response · Author response to Decision Letter 0]

13 Oct 2022

Response to the reviewers has been uploaded as a separate file.

---

## [Decision Letter · Decision Letter 1]

18 Nov 2022

PONE-D-22-10525R1How reliable is the asset score in measuring socioeconomic status? Comparing asset ownership reported by male and female heads of householdsPLOS ONE

Dear Dr. Khan,

Thank you for submitting your manuscript to PLOS ONE. After careful consideration, we feel that it has merit but does not fully meet PLOS ONE’s publication criteria as it currently stands. Therefore, we invite you to submit a revised version of the manuscript that addresses the points raised during the review process.

We look forward to receiving your revised manuscript.

Kind regards,

Rajesh Raushan, PhD

Academic Editor

PLOS ONE

Journal Requirements:

Reviewers' comments:

Reviewer's Responses to Questions

**Comments to the Author**

1. If the authors have adequately addressed your comments raised in a previous round of review and you feel that this manuscript is now acceptable for publication, you may indicate that here to bypass the “Comments to the Author” section, enter your conflict of interest statement in the “Confidential to Editor” section, and submit your "Accept" recommendation.

Reviewer #1: All comments have been addressed

Reviewer #2: All comments have been addressed

2. Is the manuscript technically sound, and do the data support the conclusions?

Reviewer #1: Yes

Reviewer #2: Yes

3. Has the statistical analysis been performed appropriately and rigorously? 

Reviewer #1: I Don't Know

Reviewer #2: Yes

4. Have the authors made all data underlying the findings in their manuscript fully available?

Reviewer #1: Yes

Reviewer #2: No

5. Is the manuscript presented in an intelligible fashion and written in standard English?

Reviewer #1: Yes

Reviewer #2: Yes

6. Review Comments to the Author

Reviewer #1: Thank you for the changes to the paper. It reads much better now and will be more accessible for non-economists. It is a crucial issue for health inequalities and needs to be disseminated widely in public health and international health circles.

On page six you state “Resorting to strategic responses in reporting asset ownership will be minimal, if any, as the link between asset ownership and means-testing programs is not clear cut.” I think you should be clearer that this is your assumption going in. I’m not sure your findings continue to support this assumption. Especially given your final paragraph about subjective social status. Adler’s ladder has had a considerable amount of subsequent research that refutes its original assumption that the subjective position was based only on the 3 domains listed in the original question wording: income, education and occupational prestige. If you are going to use the ladder choose your anchoring statements carefully – there are quite a few options in the subsequent literature. In reality, there are multiple interacting types of social position that usually are all affecting health at the same time: subjective, objective, and area-based. The correlation between them is surprisingly low.

If the “geocodes” for the data was removed, how was urban, rural, and semi-rural determined? England uses population density while other countries use land use measures such as proportion of land under agriculture.

I’m presuming that the majority of the population follows one of the Abrahamic religions so your measure of religiosity is supported as all 3 of those religions encourage people to be honest.

Your discussion seems to be more like further results. Is there no other literature to compare with your findings?

Thank you for the new Table 1. But I calculate the average number of children under 5 Total should be 2.65 rather than 2.85.

People outside of economics will not be familiar with the distinction between assets and amenities. In public health these would not necessarily be distinguished so a few sentences in the methods section would be helpful.

Reviewer #2: I believe the authors have adequately addressed all of my concerns from the previous round of reviews. The authors have provided a clear rationale for their methodological decisions, and clearly delineated how they could interpret their results.

7. PLOS authors have the option to publish the peer review history of their article (what does this mean?). If published, this will include your full peer review and any attached files.

Reviewer #1: **Yes: **Professor Shona Kelly

Reviewer #2: **Yes: **Klaus E. Cavalhieri, Ph.D.

---

## [Author Response · Author response to Decision Letter 1]

22 Nov 2022

A document on "response to reviewers" has been uploaded.

---

## [Editor Report · Decision Letter 2]

12 Dec 2022

How reliable is the asset score in measuring socioeconomic status? Comparing asset ownership reported by male and female heads of households

PONE-D-22-10525R2

Dear Dr. Khan,

We’re pleased to inform you that your manuscript has been judged scientifically suitable for publication and will be formally accepted for publication once it meets all outstanding technical requirements.

Kind regards,

Rajesh Raushan, PhD

Academic Editor

PLOS ONE

Additional Editor Comments (optional):

The manuscript can go for the publication after completing the requirements and technicalities of the journal.

Reviewers' comments:

None

---

## [Editor Report · Acceptance letter]

15 Feb 2023

PONE-D-22-10525R2 

How reliable is the asset score in measuring socioeconomic status? Comparing asset ownership reported by male and female heads of households 

Dear Dr. Khan:

I'm pleased to inform you that your manuscript has been deemed suitable for publication in PLOS ONE. Congratulations! Your manuscript is now with our production department. 

Kind regards, 

on behalf of

Dr. Rajesh Raushan 

Academic Editor

PLOS ONE